# Spectroscopic-network-assisted precision spectroscopy and its application to water

Roland Tóbiás [1], Tibor Furtenbacher [1], Irén Simkó [1], Attila G. Császár [1✉], Meissa L. Diouf[2], Frank M. J. Cozijn [2], Joey M. A. Staa [2], Edcel J. Salumbides [2] & Wim Ubachs [2✉]

Frequency combs and cavity-enhanced optical techniques have revolutionized molecular spectroscopy: their combination allows recording saturated Doppler-free lines with ultrahigh precision. Network theory, based on the generalized Ritz principle, offers a powerful tool for the intelligent design and validation of such precision-spectroscopy experiments and the subsequent derivation of accurate energy differences. As a proof of concept, 156 carefully-selected near-infrared transitions are detected for $H_2^{16}O$, a benchmark system of molecular spectroscopy, at kHz accuracy. These measurements, augmented with 28 extremely-accurate literature lines to ensure overall connectivity, allow the precise determination of the lowest *ortho*-$H_2^{16}O$ energy, now set at 23.794 361 22(25) cm$^{-1}$, and 160 energy levels with similarly high accuracy. Based on the limited number of observed transitions, 1219 calibration-quality lines are obtained in a wide wavenumber interval, which can be used to improve spectroscopic databases and applied to frequency metrology, astrophysics, atmospheric sensing, and combustion chemistry.

[1] ELTE Eötvös Loránd University and MTA-ELTE Complex Chemical Systems Research Group, Laboratory of Molecular Structure and Dynamics, Institute of Chemistry, Pázmány Péter sétány 1/A, 1117 Budapest, Hungary. [2] Department of Physics and Astronomy, LaserLaB, Vrije Universiteit, De Boelelaan 1081, 1081 HV Amsterdam, The Netherlands. ✉email: csaszarag@caesar.elte.hu; w.m.g.ubachs@vu.nl

Comprehensive spectroscopic information about small gas-phase molecules is indispensable for the characterization of various natural and artificial environments. To serve the large community of users of spectroscopic results, the data have been deposited in a number of validated, annotated, and regularly updated spectroscopic information systems, such as the HITRAN (high-resolution transmission molecular absorption) database[1]. Concurrently, the majority of wavenumber entries of line-by-line databases is based on less precise (mainly Doppler-limited) experiments, typically accurate to $10^{-3}\,\text{cm}^{-1}$. The application of precision-spectroscopy techniques, like Doppler-free cavity-enhanced saturation spectroscopy referenced to optical frequency combs, brings a new perspective to the refinement of massive amounts of transitions, entering the kHz ($10^{-7}\,\text{cm}^{-1}$) regime[2–5], and improving the accuracy of many lines and energy levels of molecular databases by orders of magnitude.

Up to now, Doppler-free precision spectroscopy has not been employed systematically to improve the quality of comprehensive spectroscopic databases, as there are experimental constraints on what we call primary line parameters (wavenumbers, Einstein $A$ coefficients, and intensities) and a single observed line may not have direct significance. For precision spectroscopy to be a useful source of line-by-line data, one needs to (a) pay particular attention to the utility of the transitions selected for measurement, and (b) know approximately the primary line parameters in advance, as searching for rovibronic lines under saturation conditions is rather time consuming.

Within the experimental constraints of the primary line parameters, one must measure lines whose detection maximizes the amount of accurate spectroscopic information gained with a minimal experimental expenditure. This goal can be achieved by viewing high-resolution spectroscopy with the generalization of the venerable Ritz combination principle[6] (see Fig. 1). The Ritz principle is arguably the most important microscopic law in spectroscopy, which establishes the connection between a transition and its lower and upper energy levels (case I of Fig. 1). Beyond its traditional, but rather limited uses (cases I and II of Fig. 1), the Ritz principle can be extended to an arbitrary number of connected spectral lines. This extension allows the definition of paths (case III of Fig. 1) and cycles (case IV of Fig. 1). Paths secure the formation of energy differences for any pair of underlying energy levels, while cycles can be used to confirm the accuracy of the associated transitions[7]. To increase the number of accurately known energy differences, one must ensure that transitions having minuscule uncertainties are connected, that is, their paths (and cycles) are not broken with inaccurate lines. Paths and cycles have been used implicitly by several spectroscopic protocols[8–14] based on least-squares inversion of transition wavenumbers to rovibronic energy values. Nevertheless, it should be noted that the methods described in refs. [8–14] do not decompose the input dataset into paths and cycles; thus, they cannot reveal which lines cause the inaccuracy of a particular energy value.

Building the list of target lines, forming accurate paths and cycles after the measurements, necessitates the use of elements of network theory[15] and the concept of spectroscopic networks[16,17], the most general extension of the Ritz principle. In spectroscopic networks, energy levels are the vertices (nodes) and transitions are the edges (links). Among other advanced computer science approaches applied to high-resolution spectroscopy (e.g., genetic algorithms[18–20] and machine learning involving artificial neural networks[21]), network theory and its sophisticated polynomial algorithms provide interesting and highly useful tools to exploit all the spectroscopic information coded in the connections of rovibronic built via experiments lines[17].

An essential property of spectroscopic networks is that the vertex degrees (edge counts of the individual nodes) follow an inverse-power-like (heavy-tailed or quasi scale-free[15]) distribution[16]. This property implies the presence of hubs (high-degree nodes) among the rovibronic energy levels. Decreasing the uncertainties with which we know the energies of hubs by precision-spectroscopy experiments is highly beneficial as hubs are the lower states of a large number of observable lines.

Since water is an important benchmark system of high-resolution molecular spectroscopy (e.g., it is molecule no. 1 in HITRAN[1]), its main isotopologue $H_2^{16}O$ was subject to detailed investigation in the literature. Over the past 100 years, some 200,000 rovibrational lines have been recorded for $H_2^{16}O$ (linking nearly 20,000 energy levels[22,23]) to probe water in the interstellar medium[24], in exoplanets[25], on the Sun[26], and in the Earth's atmosphere,[27] including its greenhouse effect[28,29]. The rovibrational states of $H_2^{16}O$ are separated into two subsets, corresponding to the *ortho* and *para* nuclear-spin isomers. As of today, no *ortho–para* transitions have been observed in water vapor[30], and thus the energy separation of the *ortho* and *para* states is not known precisely. The list of spectral lines of $H_2^{16}O$ measured at high accuracy ($10^{-7}\,\text{cm}^{-1}$) is short, and derives mainly from microwave and THz experiments[31–41], except lines from two near-infrared Doppler-free laser studies[42,43].

In this study, noise-immune cavity-enhanced optical heterodyne molecular spectroscopy (NICE-OHMS) measurements are performed for $H_2^{16}O$, yielding 156 rovibrational transitions under Doppler-free conditions in the near infrared (above 7000 cm$^{-1}$). The lines to be measured are selected via a combined network-theoretical and experimental approach (spectroscopic-network-assisted precision spectroscopy, SNAPS), aided by the availability of experimental[44] and first-principles[45] linelists. The extremely accurate lines observed help to (a) provide an extremely accurate estimate of the lowest *ortho*-$H_2^{16}O$ energy value; (b) improve the accuracy of the energies of numerous hubs, (c) assess the accuracy of the present as well as previous literature results, mostly in the THz region and within the ground vibrational state; (d) construct the largest experimental-quality linelist in the literature with some $10^{-7}\,\text{cm}^{-1}$ uncertainty; and (e) analyze subtle systematic effects, mainly ignored in previous studies, like the pressure dependence of the frequencies of Doppler-free lines.

## Results and discussion

**The SNAPS approach.** The main result of this paper is the SNAPS approach, which is a universal, versatile, and flexible algorithm, designed for all measurement techniques and molecules where the rovibrational lines are resolved individually (with their frequencies measured at extreme accuracy). The SNAPS procedure (a) starts with the selection of the most useful set of target transitions allowed by the range of primary line parameters, (b) continues with the measurement of the target lines, (c) supports cycle-based validation[46] of the accuracy of a large number of detected lines, and (d) allows the transfer of the high experimental accuracy to the derived energy values and predicted line positions. Details are presented in the Methods section. Although the SNAPS protocol strongly relies on network theory, it can be deemed as a black-box-type strategy: its output (the target linelist and the sets of generated paths and cycles) can be understood merely via the extended Ritz principle.

Taking into account the experimental constraints on the primary line parameters and employing 28 lines from the literature[31,37,39,42], 156 rovibrational transitions of $H_2^{16}O$ have been selected with the SNAPS approach and observed with the NICE-OHMS apparatus. Figures 2 and 3 give an overview of the present and former [(sub-)kHz accuracy] measurements for *ortho-* and *para*-$H_2^{16}O$, respectively. As

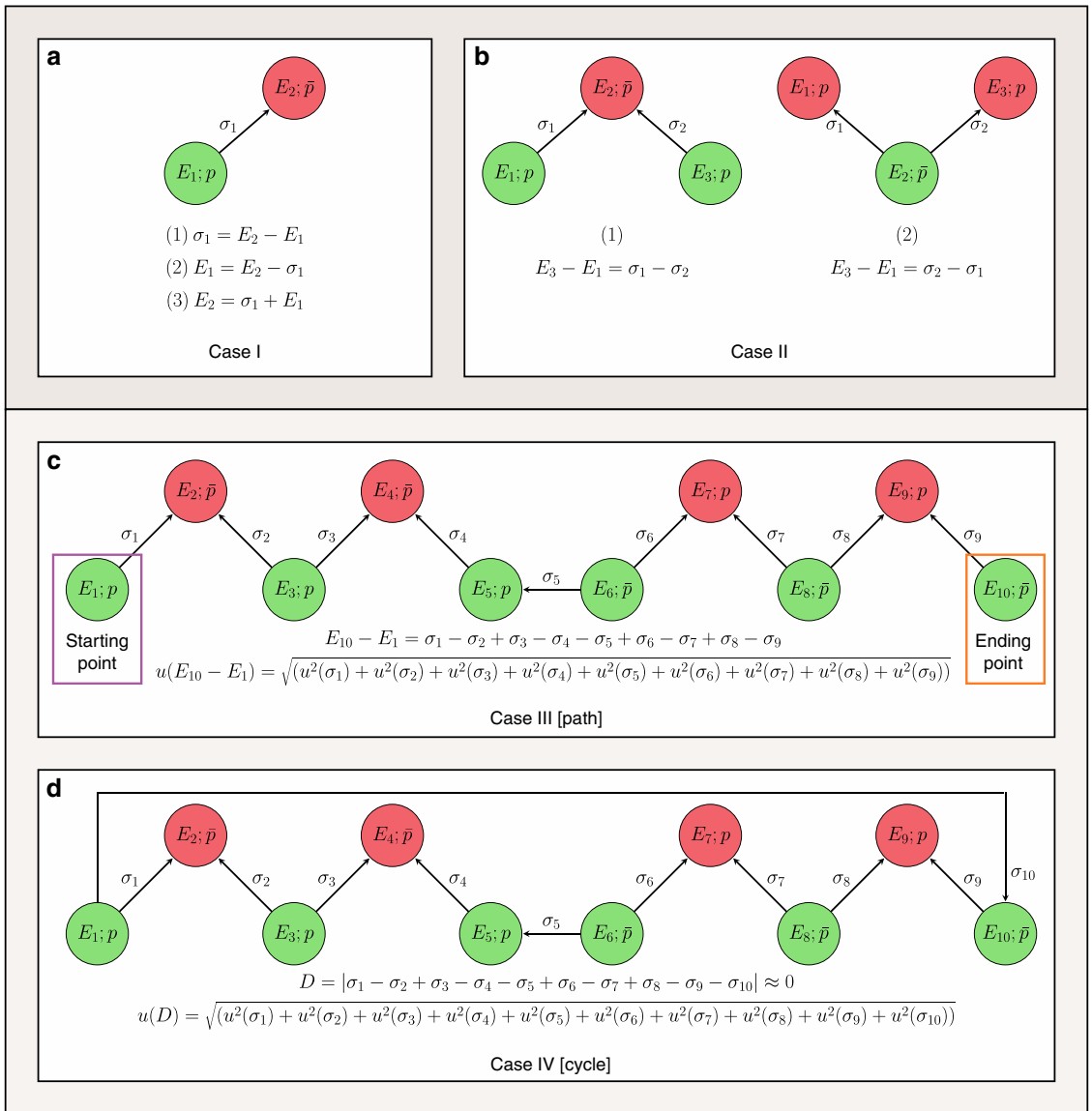

**Fig. 1 Graphical illustration of the spectroscopic utilizations of the Ritz principle.** In this figure, nodes and edges are energy levels and (one-photon, dipole-moment-allowed) transitions with given energy values ($E_i$) and wavenumbers ($\sigma_i$), respectively. The parity ($p$: even/odd; $\bar{p}$: odd/even) is displayed for each rovibronic state. Cases I (**a**) and II (**b**): traditional applications of the Ritz principle, usually called combination differences, where case I/1 corresponds to the definition of the Ritz principle. Generalization of the Ritz principle leads to paths (sequences of connected, unrepeated transitions and energy levels—case III, **c**) and cycles (series of connected transitions and energy levels, where each rovibronic level is connected to two other energy levels—case IV, **d**). On a path, one can predict the energy differences of all its pairs of states, applying the Ritz principle in a successive way. If the starting point in case III is the lowest-energy state reachable from the ending point via paths, then $E_{10} - E_1$ is referred to as the relative energy of the ending point. In the special case when the starting point is the rovibronic ground state of the molecule, $E_1 = 0$ by definition, and $E_{10} - E_1$ provides the energy value of the ending point. Based on the $u(\sigma_i)$ wavenumber uncertainties, the accuracy of the predicted energy differences can also be estimated from the law of uncertainty propagation (assuming uncorrelated measurement errors with zero expected values). Cycles reflect the internal accuracy of their lines: in favorable cases, $D \leq t_{crit} u(D)$, where $D$ is the discrepancy (absolute signed sum of the related wavenumbers) of the cycle with its $u(D)$ uncertainty, and $t_{crit} \approx 2$ (see Supplementary Note 1). If the states denoted with the same (green or red) color have the same vibronic parent, it can be recognized that, due to the rotational selection rule, the jumps altering the vibronic states produce two separate subsets of levels for the green states (and also for the red ones) with parities $p$ and $\bar{p}$. These subsets can be linked only by pure rotational lines with $\sigma_5$ and $\sigma_{10}$.

obvious from Figs. 2 and 3, the previous experiments form several isolated single paths (islands), while the inclusion of the NICE-OHMS lines makes both the *ortho* and the *para* lines connected, thus increasing considerably the overall utility of even those lines taken from the literature.

The remaining part of this section is divided into two majorsections. First, the important experimental results are presented. Then, it is shown how the transitions detected in a limited wavenumber range can be utilized to gain information for other spectral regions.

**NICE-OHMS precision spectroscopy of H$_2$$^{16}$O**. The experiments utilized a NICE-OHMS apparatus[47,48] (see Methods) and we recorded 156 absorption lines of H$_2$$^{16}$O in saturation. The wavenumber coverage of the measurements is limited, mainly by the

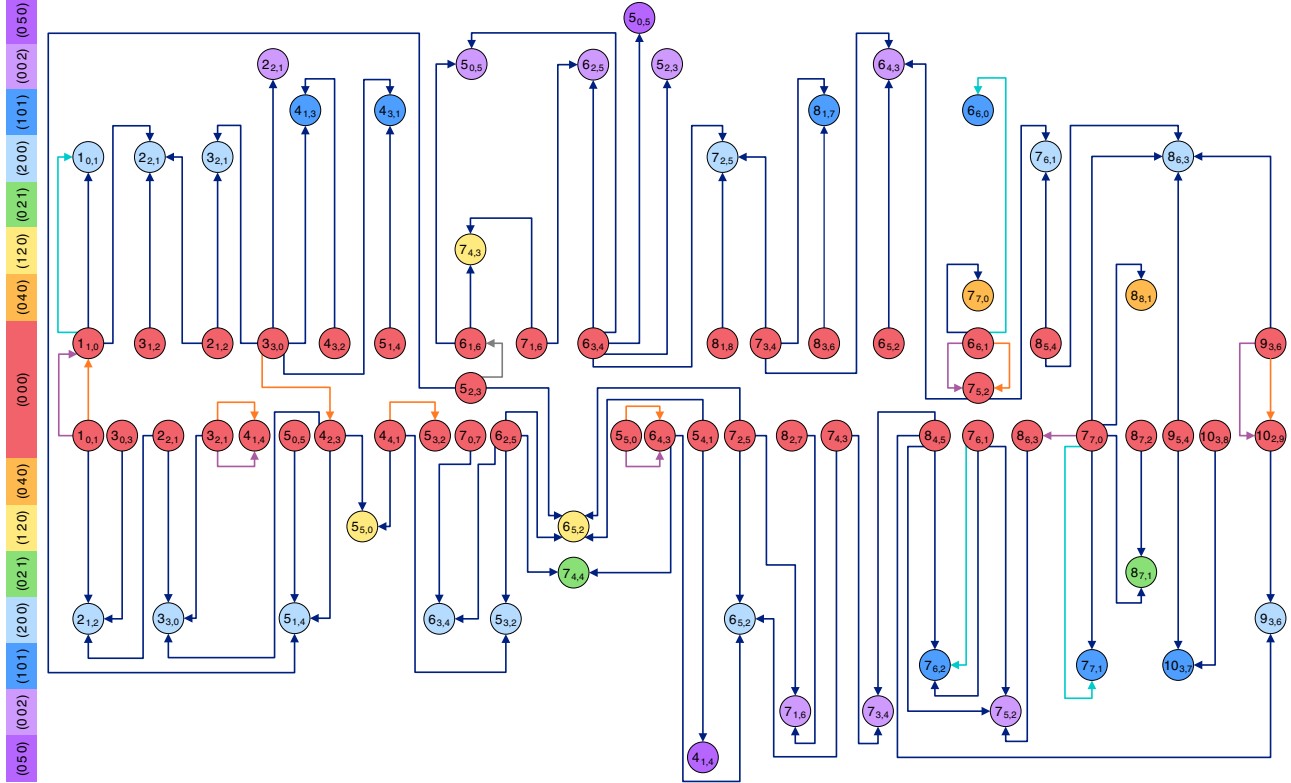

**Fig. 2 Pictorial representation of all the precision measurements for *ortho*-H$_2$$^{16}$O.** The rovibrational states are labeled with $(v_1\ v_2\ v_3)J_{K_a,K_c}$, where $(v_1\ v_2\ v_3)$ contains the normal-mode quantum numbers (following the Mulliken convention[69]) and $J_{K_a,K_c}$ corresponds to the asymmetric-top quantum numbers[70]. The *ortho* energy levels (characterized with odd $v_3 + K_a + K_c$ values and denoted with circles) are placed palindromically in increasing (upper half) and decreasing (lower half) energetic order of their vibrational parents and distinguished with different colors for improved transparency. The $J_{K_a,K_c}$ rotational labels are indicated individually for each rovibrational state, while the $(v_1\ v_2\ v_3)$ vibrational assignments[69] are marked in the left-side color legend. The eight vibrational states of the figure correspond to the $P = 0$ [(0 0 0)], $P = 4$ [(0 4 0), (1 2 0), (0 2 1), (2 0 0), (1 0 1), and (0 0 2)] and $P = 5$ [(0 5 0)] polyads, where $P = 2v_1 + v_2 + 2v_3$ is the polyad number. Transitions with blue arrows are results of the present NICE-OHMS experiments, while those with gray, orange, purple, and cyan colors are taken from refs. [31,37,39,42], respectively. Source data are provided as a Source Data file.

highly reflective cavity mirrors, to 7000–7350 cm$^{-1}$. The accessible window of transition dipole moments, represented by the Einstein $A$ coefficients, falls in the interval of $10^{-4}$–$10^2$ s$^{-1}$. Combined with the population distribution at room temperature, this factor leads to the intensity interval of $10^{-26}$–$10^{-20}$ cm molecule$^{-1}$.

All experiments were performed through saturation spectroscopy to obtain Doppler-free transitions, typically resulting in linewidths on the order of 100 kHz (half width at half maximum (HWHM)). The vast increase in resolution compared to Doppler-broadened techniques allows to resolve closely spaced transitions, such as the $(0\ 4\ 0)8_{8,1/0} \leftarrow (0\ 0\ 0)7_{7,0/1}$ *ortho–para* doublet shown in Fig. 4a, which would otherwise be unresolvable. By comparing the spectroscopy laser to a frequency comb laser, via a beat-note measurement, the spectra are related to an absolute frequency scale of sub-kHz accuracy, providing individual uncertainties of a few kHz for the line centers.

Systematic studies of power-broadening and pressure-shift effects were also undertaken. Some typical results are presented in Fig. 4b and c. The line center frequencies were extrapolated to zero pressure to correct them for pressure shifts (see also Methods). An additional small effect of line broadening was found as a result of hyperfine structure in the case of *ortho*-H$_2$$^{16}$O (see also Fig. 4b and the Methods section). Based on a combined treatment of statistical and systematic errors, the transition frequencies were associated with individual uncertainties for each line (see also Methods).

The most precise line [$(2\ 0\ 0)1_{0,1} \leftarrow (0\ 0\ 0)1_{1,0}$] reported in a near-infrared study of Kassi et al.[42] with 3 kHz accuracy was used

for to verify our data treatment. It was remeasured with 1.8 kHz uncertainty, the two measurements agree within their uncertainty limits. Beyond this line repeated measurements were performed for three more lines from ref. [42] (see Figs. 2 and 3). In these cases, the NICE-OHMS results proved to be a factor of four more accurate than their previous determinations[42].

In addition to these repeated measurements forming trivial (two-membered) cycles, several longer (mostly four- and six-membered) cycles were also built principally from lines of Figs. 2 and 3 to check the self-consistency of the line positions and assignments. These cycles occasionally revealed accidental mistakes made during the determination of line centers from the raw experimental data and proved that some literature lines have much larger uncertainties than indicated in the original publications. Consequently, measuring well-designed cycles is highly recommended to check the correctness and the accuracy of the recorded lines. The discrepancies of all the investigated cycles are a few times $10^{-7}$ cm$^{-1}$. This suggests an excellent internal consistency of the observed lines.

**Transfer of measurement accuracy.** It is a significant problem how to transfer the high accuracy of the NICE-OHMS measurements to that of derived energy values and wavenumbers in the best possible way. The extended Ritz principle and the SNAPS approach offer a solution here, as well. Building paths out of observed lines ensures that several, explicitly unmeasured energy differences can be calculated with definitive uncertainties (see case III of Fig. 1). Such a predicted energy difference (a) relies

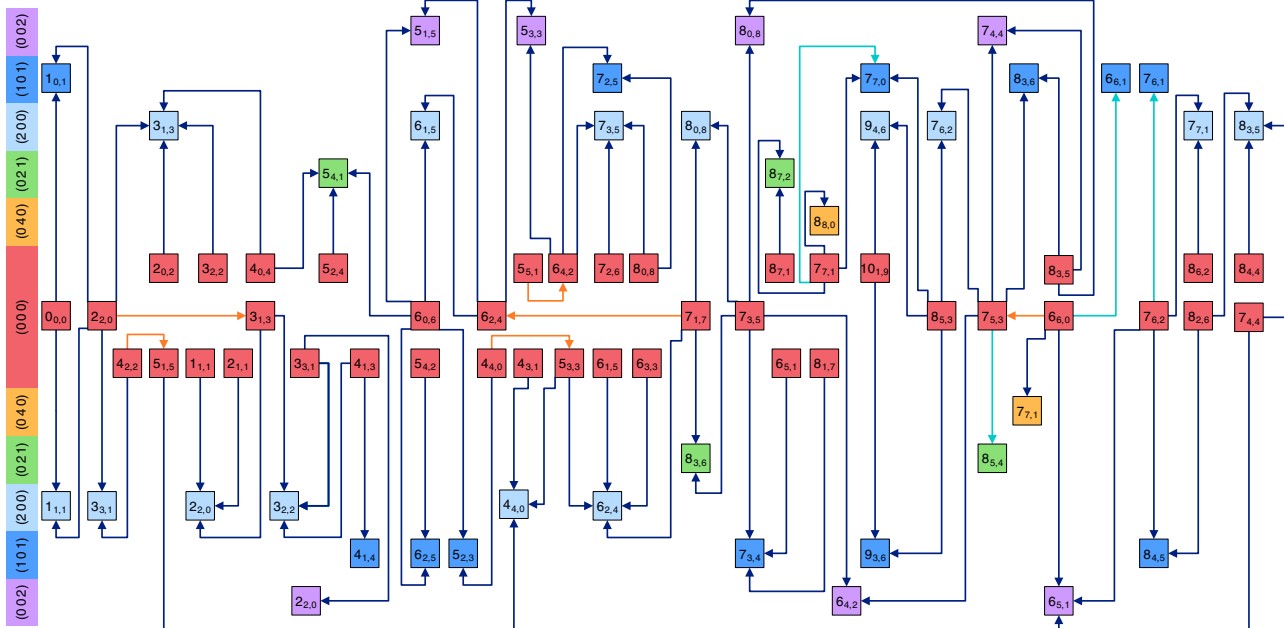

**Fig. 3 Pictorial representation of all the precision measurements for *para*-H$_2$$^{16}$O.** The *para* energy levels (characterized with even $v_3 + K_a + K_c$ values and denoted with squares) are placed palindromically in increasing (upper half) and decreasing (lower half) energetic order of their vibrational parents and distinguished with different colors for improved transparency. The $J_{K_a,K_c}$ rotational labels[70] are indicated individually for each rovibrational state, while the ($v_1 v_2 v_3$) vibrational assignments[69] are marked in the left-side color legend. Transitions with blue arrows are results of the present NICE-OHMS experiments, while those with orange and cyan colors are taken from refs. [37,42], respectively. Source Data are provided as a Source Data file.

only on the experimental wavenumbers and therefore does not suffer from overfitting and other numerical effects, (b) may have an accuracy characteristic of the NICE-OHMS setup, (c) is not restricted to the original experimental wavenumber range, and (d) may be exempt from some systematic errors, which may cancel each other due to subtractions in its defining equation. As a result, these energy-difference estimates can supersede those obtained from effective Hamiltonian models or direct results of lower-resolution techniques. In what follows, three relevant applications are presented, where the accuracy of the NICE-OHMS technique could be transferred to energy differences.

**Highly accurate relative energy values**. Since the high-precision *ortho* (Fig. 2) and *para* (Fig. 3) lines of H$_2$$^{16}$O form two connected sets, we are able to derive relative energy values for all the available *ortho* and *para* states, starting from the (0 0 0)1$_{0,1}$ and (0 0 0)0$_{0,0}$ energy levels, respectively. Thus, one can construct paths starting at (0 0 0)1$_{0,1}$ or (0 0 0)0$_{0,0}$ and explore all of the rovibrational states of Figs. 2 and 3. A typical path is depicted in Fig. 5a, which defines the (relative) energy value of the (0 0 0)1$_{1,1}$ state.

During the SNAPS procedure, one of the principal objectives is to obtain accurate relative energies for the most important hubs of H$_2$$^{16}$O. Among the 185 hubs (top 1% of the known H$_2$$^{16}$O energy levels in descending order of their vertex degrees) of the IUPAC (International Union of Pure and Applied Chemistry) database of H$_2$$^{16}$O[22], 153 and 32 lie on the (0 0 0) and (0 1 0) vibrational states with $J \leq 14$ and $J \leq 9$, respectively. As the (0 0 0) hubs cover virtually all the (0 0 0) energy levels for $J \leq 8$ [only (0 0 0)8$_{8,0}$ is missing from the hub list], we decided to redefine the relative energy values of all the (0 0 0)$J_{K_a,K_c}$ states up to $J = 8$ [apart from (0 0 0)8$_{8,0}$ and (0 0 0)8$_{8,1}$, which would have required measuring a few too weak target lines] based on paths of Figs. 2 and 3. In the present NICE-OHMS wavenumber range, we could not find paths linking the (0 1 0) hubs to the (0 0 0)0$_{0,0}$ or (0 0 0) 1$_{0,1}$ states. Thus, no (0 1 0) relative energies could be determined during this study.

Beyond hubs, some further representative rovibrational states were also investigated within the vibrational bands attainable under the experimental conditions applied. In total, 160 relative energy values have been redetermined with an expected accuracy of some $10^{-7}$ cm$^{-1}$. Of the underlying rovibrational energy levels, 84, 15, 6, 4, 2, 19, 3, and 27 lie on the (0 0 0), (0 0 2), (0 2 1), (0 4 0), (0 5 0), (1 0 1), (1 2 0), and (2 0 0) vibrational states, respectively. Despite the fact that the NICE-OHMS measurements performed in the near-infrared region involve highly excited vibrational parents, the accuracy of the energy levels derived within the ground vibrational state is one or even two orders of magnitude better than those obtained by pure rotational measurements in the THz frequency range[49–53].

**Lowest *ortho* energy value of H$_2$$^{16}$O**. As emphasized in the Introduction, no transitions connecting *ortho* and *para* states of H$_2$$^{16}$O have been detected. This observational hiatus necessitates the use of indirect protocols to obtain the (0 0 0)1$_{0,1}$–(0 0 0)0$_{0,0}$ energy splitting, that is, the lowest *ortho* energy value of the H$_2$$^{16}$O molecule.

The best previous estimates[54–57] for the lowest *ortho* energy are accurate to $10^{-6}$–$10^{-5}$ cm$^{-1}$. This means that none of the *ortho*-H$_2$$^{16}$O energy values can have an absolute uncertainty better than some $10^{-6}$ cm$^{-1}$, even those involved in the much more accurate NICE-OHMS measurements. To improve the absolute accuracy of all the *ortho* energies, one needs to determine the lowest *ortho* energy with an uncertainty of a few $10^{-7}$ cm$^{-1}$.

The traditional way to derive the lowest *ortho* energy involves a fit of an effective Hamiltonian model to the largest number of energy differences within the ground vibrational state. To simplify the fitting procedure, we constructed artificial transitions from the vibration-mode-changing lines of Figs. 2 and 3 sharing the same upper states and calculated the underlying (0 0 0) energy differences with their uncertainties (see also cases II and III of Fig. 1). Building a training set (restricted to $J \leq 8$) from the appropriate artificial lines and pure rotational

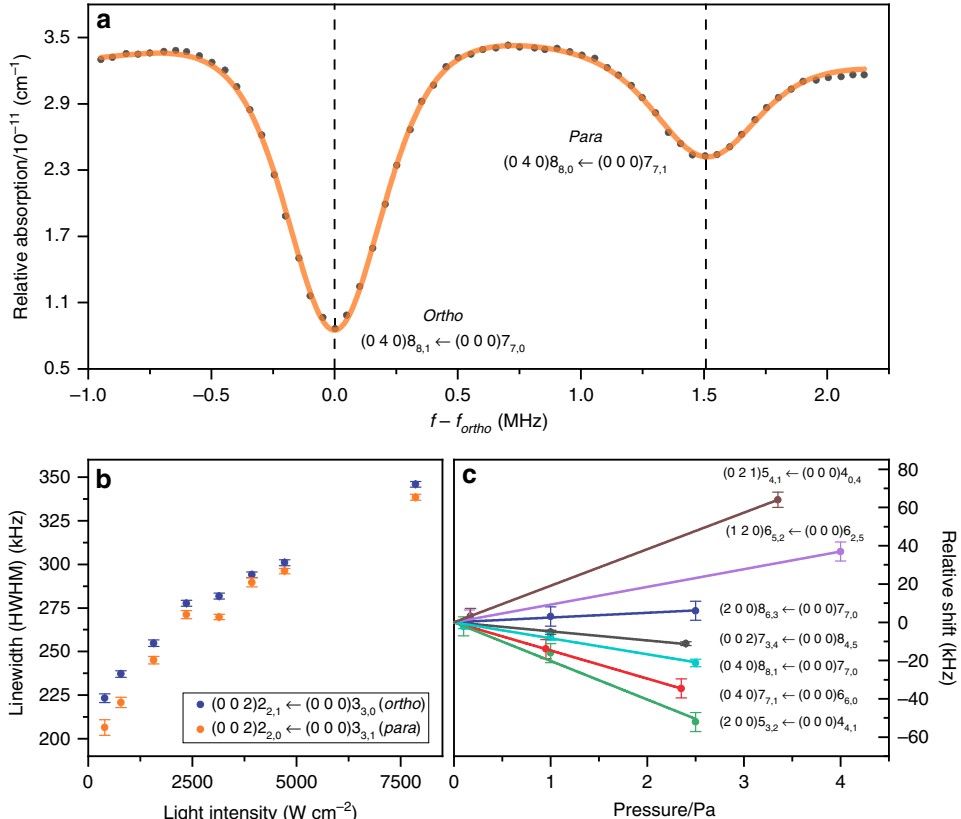

**Fig. 4 Overview of the NICE-OHMS experiments performed for $H_2{}^{16}O$. a** NICE-OHMS spectrum of the $(0\ 4\ 0)8_{8,1/0} \leftarrow (0\ 0\ 0)7_{7,0/1}$ doublet. The scanning frequency, $f$, is shifted with the line position of the *ortho* transition, $f_{ortho} = 218,444,703.636$ MHz (separated by 1.5 MHz from the *para* line). As clear from this spectral recording, the *ortho* line exhibits a three times larger intensity than its *para* counterpart, which is due to the nuclear-spin degeneracy factors and the nearly equal Einstein $A$ coefficients of the lines. Note that the assignment of this doublet is interchanged in HITRAN2016[1]. **b** Measured linewidth (half width at half maximum, HWHM) of the saturated resonance as a function of intracavity intensity for the $(0\ 0\ 2)2_{2,1/0} \leftarrow (0\ 0\ 0)3_{3,0/1}$ *ortho–para* doublet. The pressure-free wavenumbers of these two lines are determined to be 7286.715 825 626(60) (*ortho*) and 7285.044 729 204(63) cm$^{-1}$ (*para*). **c** Pressure-shift study of the saturated absorption line centers for some $H_2{}^{16}O$ lines with their assignments. The error bars represent one standard deviation. Source data are provided as a Source Data file.

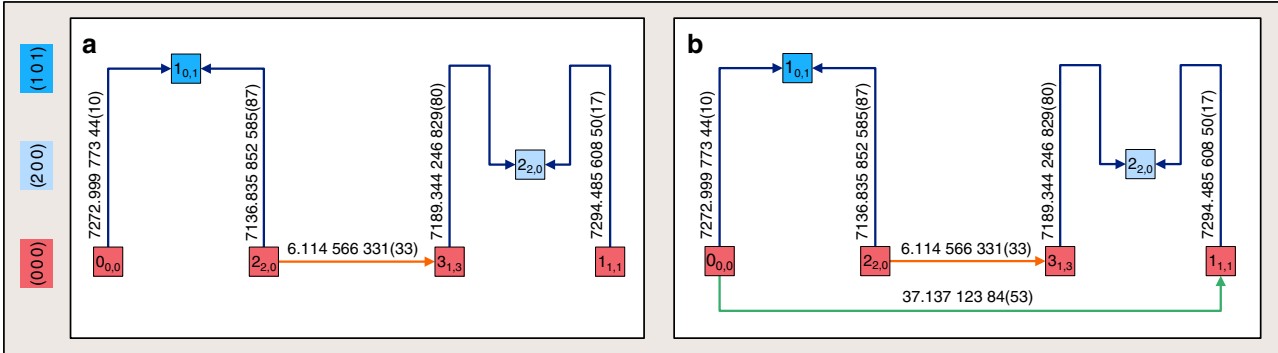

**Fig. 5 Application of the generalized Ritz principle to the $(0\ 0\ 0)1_{1,1}$ state. a**, **b** show a path ($\mathcal{P}$) and a cycle ($\mathcal{C}$) involving the $(0\ 0\ 0)1_{1,1}$ level, respectively. The $J_{K_a,K_c}$ rotational labels[70] are indicated for each rovibrational state, while the canonical $(v_1v_2v_3)$ vibrational assignments[69] are marked in the left-side color legend. Transitions with blue arrows are results of the present NICE-OHMS experiments, while those with orange and green colors are taken from refs. [37,49], respectively. Each line is connected with its wavenumber (in cm$^{-1}$) and the uncertainty of the last few wavenumber digits is within parentheses. Successive application of the Ritz principle leads to **a** the best estimate of the $(0\ 0\ 0)1_{1,1}$ energy value, 37.137 125 52(23) cm$^{-1}$, derived from $\mathcal{P}$, and **b** the discrepancy of $\mathcal{C}$, supplied with an uncertainty, 1.68(58) × 10$^{-6}$ cm$^{-1}$ (see cases III–IV of Fig. 1). The energies of the internal states [$(1\ 0\ 1)1_{0,1}$, $(0\ 0\ 0)2_{2,0}$, $(0\ 0\ 0)3_{1,3}$, and $(2\ 0\ 0)2_{2,0}$] can be similarly obtained from $\mathcal{P}$. Accepting the line uncertainties of $\mathcal{P}$, the relatively large discrepancy of $\mathcal{C}$ indicates that the $(0\ 0\ 0)1_{1,1}$ energy can be more accurately calculated from $\mathcal{P}$ than from the direct link taken from ref. [49]. Source data are provided as a Source Data file.

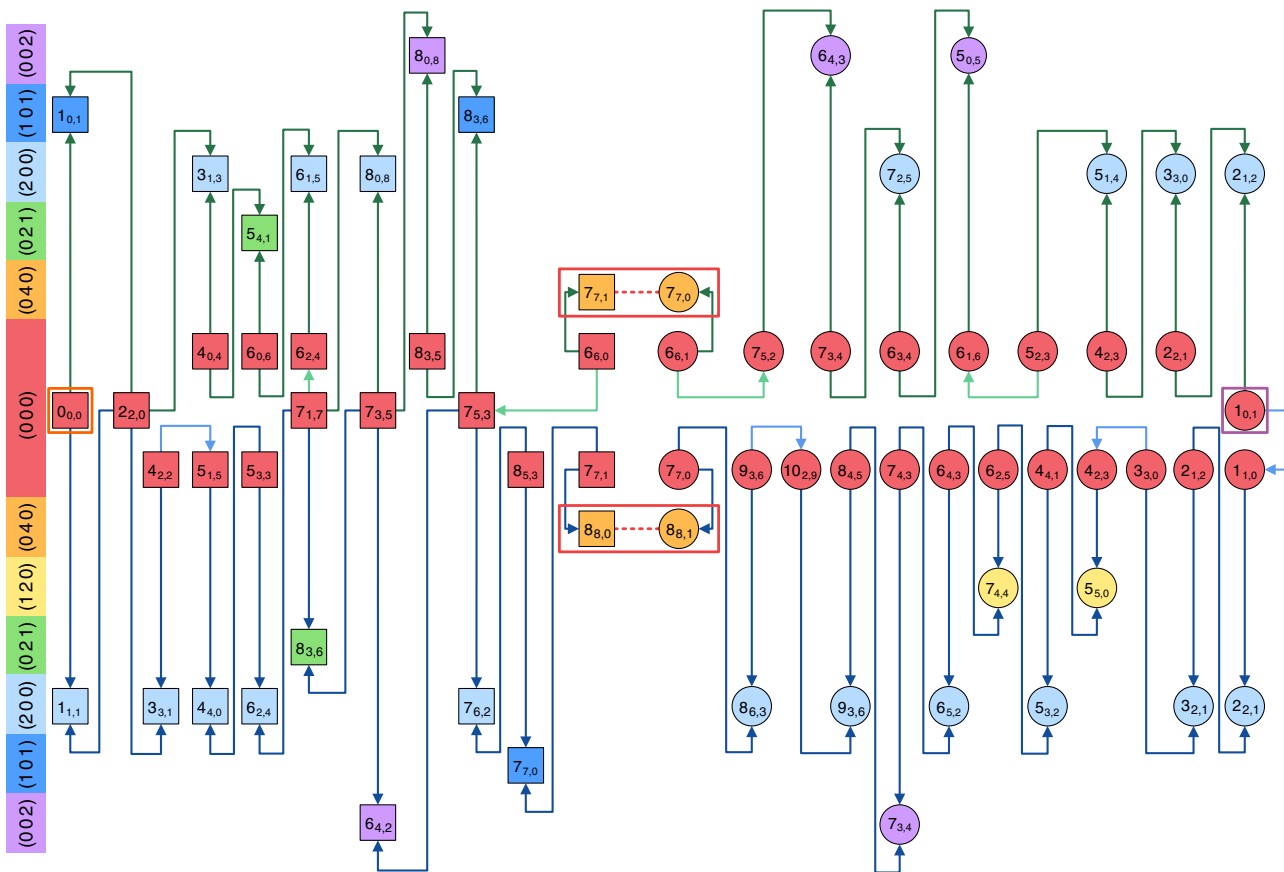

**Fig. 6 Two line-disjoint paths used for the derivation of the lowest *ortho*-H$_2^{16}$O energy.** The $J_{K_a,K_c}$ rotational labels[70] are indicated individually for each rovibrational state, while the canonical $(v_1\,v_2\,v_3)$ vibrational assignments[69] are marked in the left-side color legend. Transitions of paths #1 and #2 are highlighted with green and blue arrows, respectively, where the *ortho* and *para* energy levels are represented with circles and squares, respectively. The starting and ending points of these paths, $(0\,0\,0)0_{0,0}$ and $(0\,0\,0)1_{0,1}$, are highlighted by orange and purple squares, respectively. Common states of paths #1 and #2 are placed in the middle row of the figure. Transitions with darker colored arrows are results of the present study, while the rotational lines with lighter colored connectors are taken from refs. [31,37,39]. The dashed red lines surrounded by red boxes represent virtual transitions whose wavenumbers correspond to small *ortho–para* energy splittings [2.99(17) × 10$^{-6}$ and 4.26(18) × 10$^{-6}$ cm$^{-1}$ for $(0\,4\,0)8_{8,0/1}$ and $(0\,4\,0)7_{7,0/1}$, respectively] derived from nuclear-motion (GENIUSH) computations[65,66] (see also Source Data and Supplementary Note 4). Relying on the Ritz principle and the law of uncertainty propagation (see formulas of case III in Fig. 1) as well as the wavenumbers and uncertainties of the virtual transitions, paths #1 and #2 give the estimated values of 23.794 361 54(59) and 23.794 361 41(71) cm$^{-1}$ for the lowest *ortho* energy, respectively. For details on the generation and evaluation of these two paths, see Methods. Source data are provided as a Source Data file.

transitions of refs. [31,37,39], a weighted least-squares fit was performed with a 14th-order Watson-type Hamiltonian. This high order is required due to the considerable accuracy of the lines included in the training set. To make an external validation for the fitting, we excluded all those lines incident to levels $(0\,0\,0)1_{1,1}$, $(0\,0\,0)2_{0,2}$, $(0\,0\,0)2_{1,1}$, and $(0\,0\,0)2_{2,0}$ from the training set. The fit with this effective Hamiltonian leads to the lowest *ortho* energy estimate of 23.794 361 22(25) cm$^{-1}$, while perfectly reproducing the energies of the $(0\,0\,0)1_{1,1}$, $(0\,0\,0)2_{0,2}$, $(0\,0\,0)2_{1,1}$, and $(0\,0\,0)2_{2,0}$ states within their stated uncertainties. Further details on the effective Hamiltonian modeling are given in Supplementary Note 2.

An independent, network-based way to determine the lowest *ortho* energy is to seek and measure all the shortest possible line-disjoint paths from $(0\,0\,0)0_{0,0}$ to $(0\,0\,0)1_{0,1}$, involving accurate virtual transitions that connect closely spaced $(v_1\,v_2\,v_3)J_{J,0/1}$*ortho–para* state pairs with sufficiently high $J$ values (for details, see Methods). This protocol yielded only two such paths (see Fig. 6) within the experimental restrictions, upon which two independent determinations [23.794 361 54(59) and 23.794 361 41(71) cm$^{-1}$] are obtained for the lowest *ortho* energy. These data are in full accord with our effective Hamiltonian value.

Based on a careful consideration of all the results, the estimate of our effective Hamiltonian model, 23.794 361 22(25) cm$^{-1}$, is recommended as the new reference value for the lowest *ortho* energy. A comparison of the present lowest *ortho* energy determinations with results of previous effective Hamiltonian models[49,58–62,54,55,56] is displayed in Fig. 7 (for comments, see Supplementary Note 5).

Finally, it must be emphasized that the traversal of the $(0\,0\,0)$ rotational energy levels via up and down jumps involving several vibrational bands (see Fig. 6) is a fully novel approach, inspired by the network-theoretical view of high-resolution spectroscopy. Construction of paths with similar complexity would be impossible via conventional spectroscopic tools (combination difference relations or effective Hamiltonian models).

**High-accuracy predicted linelist**. The list of lines predicted from the paths of Figs. 2 and 3 may act as frequency standards[63] over a wide range, helping future high-resolution experiments. This is especially true for those 30 pairs of predicted transitions whose separation is <0.01 cm$^{-1}$, which cannot easily be resolved via Doppler-broadened measurements. These highly accurate

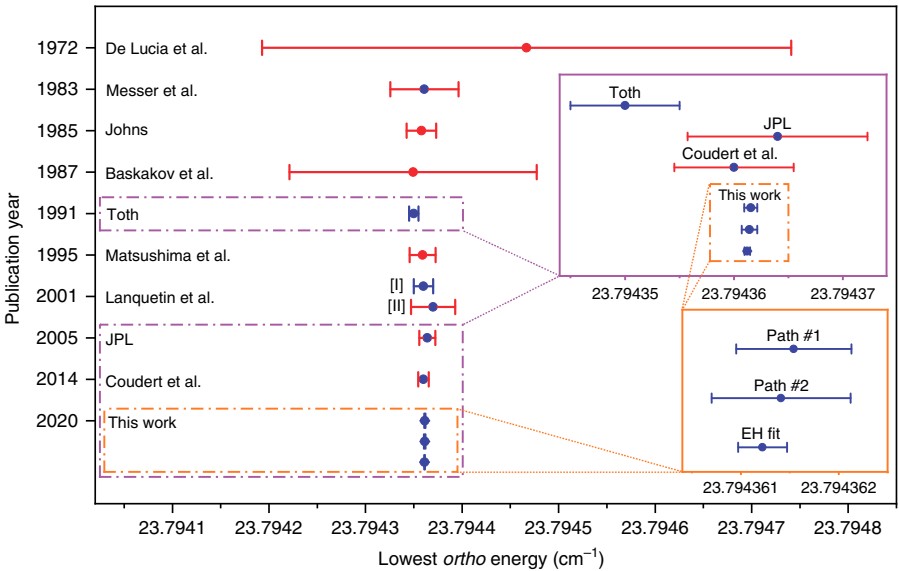

**Fig. 7 Overview of selected determinations of the lowest *ortho* energy of H$_2$$^{16}$O.** On the vertical axis, the publication year is provided for the present and the literature[49,58–62,54,55,56] determinations. In ref. [54], two different estimates ([I] and [II] in this figure) are reported. Data from Toth[62], JPL[55], and Coudert et al.[56], as well as from this work are displayed in the purple inset. Our predictions [paths #1 and #2: paths of Fig. 6; EH fit: ground vibrational-state effective Hamiltonian fit (see the main text and Supplementary Note 2)] are repeated in the orange inset for improved transparency. Blue dots are published lowest *ortho* energy values, while the red ones have been calculated during the present study from the effective Hamiltonian parameters of the original publications. Similarly, blue error bars illustrate originally reported uncertainties, while the red ones are our estimates defined in Supplementary Eq. 12 and analyzed in Supplementary Note 5. Source data are provided as a Source Data file.

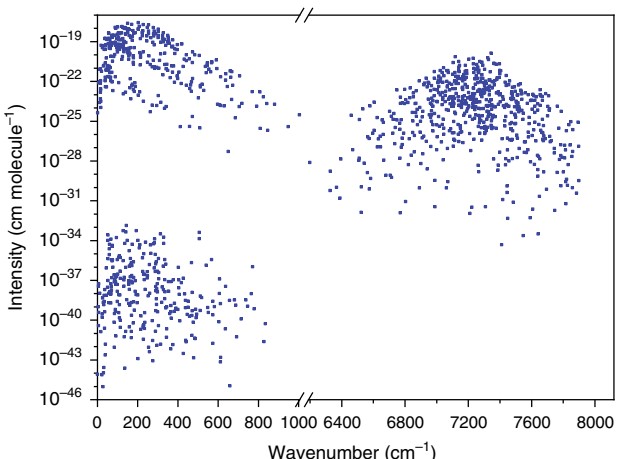

**Fig. 8 High-accuracy linelist of H$_2$$^{16}$O generated from paths of Figs. 2 and 3 (see also Supplementary Note 6).** In addition to the transition wavenumbers, 296 K absorption intensities[64] are also presented on the vertical axis. The data points in the left lower quadrant correspond to low-intensity, vibration-mode-changing transitions. While the directly measured lines are constrained to the interval 7000–7350 cm$^{-1}$, the predicted transitions resulting from lines of Figs. 2 and 3 extend between 0–1005 and 6198–8114 cm$^{-1}$. Source data are provided as a Source Data file.

calculated lines, having two or three orders of magnitude smaller uncertainties than their former determinations, give us the opportunity to validate less accurate measurements and reveal individual and systematic (e.g., pressure-shift) effects in their line positions. Such an examination is mostly important for the THz regime, where the usually applied experimental techniques provide a few times 10$^{-6}$ cm$^{-1}$ accuracy for the detected transitions. The complete set of predicted lines is presented in Fig. 8. An assessment of the literature data sources corresponding to our

predicted lines is provided in Supplementary Note 6 and Supplementary Fig. 3.

As illustrated here for H$_2$$^{16}$O, the SNAPS method performs excellently; therefore, this approach is highly recommended to enhance our understanding of the spectroscopy of important molecular species, similar to those in HITRAN. It is expected that SNAPS, once applied to the majority of molecules of interest (note that the latest HITRAN version[1] contains rovibronic lines for 49 molecules), will improve significantly the accuracy and utility of spectroscopic databases.

## Methods

**Theory.** The SNAPS procedure can be divided into four major phases (Fig. 9): [I] preparation phase, where the SNAPS input is created, [II] selection phase, where the full set of possible target lines is generated, [III] measurement phase, where the list of target lines is reduced and then detected with a precision-spectroscopy setup, and [IV] evaluation phase, where the final paths and cycles are obtained and assessed. Phases II and IV are based on elements of network theory, while phase III is where the theoretical knowledge is turned into experimental information. The network-theoretical notions and methods utilized here are summarized in Supplementary Note 1 and illustrated in Supplementary Fig. 1.

In phase I (preparation), five input units are introduced. In the reference dataset, those (highly accurate) literature lines are placed which one desires to build into paths and cycles. The template dataset comprises transitions with their estimated (experimental or calculated) primary line parameters and wavenumber uncertainties. The experimental constraints (the wavenumber window of [$\sigma_{min}$, $\sigma_{max}$], the intensity interval of [$S_{min}$, $S_{max}$], and the Einstein $A$ coefficient range of [$A_{min}$, $A_{max}$]) imposed upon the template lines are characteristic of the precision-spectroscopy technique applied. If one opts for the determination of the energy differences of some state pairs, the list of these state pairs (state duals) should be provided. For these state duals, all the available line-disjoint paths will be found. The extracted dataset is initially empty, which will be filled with extracted (newly measured and resolved) lines provided by the experimental method chosen.

As the first stage of phase II (selection), a design network is constructed from (a) all the reference and extracted transitions, and (b) those template lines that meet the experimental constraints and do not have counterparts with the same assignment in the extracted dataset. Then, shortest-path searches are performed by running Dijkstra's algorithm[15] to explore the structure of the design network. For these searches, the weights of the lines are specified as follows: (a) zero weights are assigned to all the transitions of the reference and extracted dataset (i.e., they are forced into the shortest paths to reduce the experimental cost), and (b) weights of the template lines are represented by the squared wavenumber uncertainties.

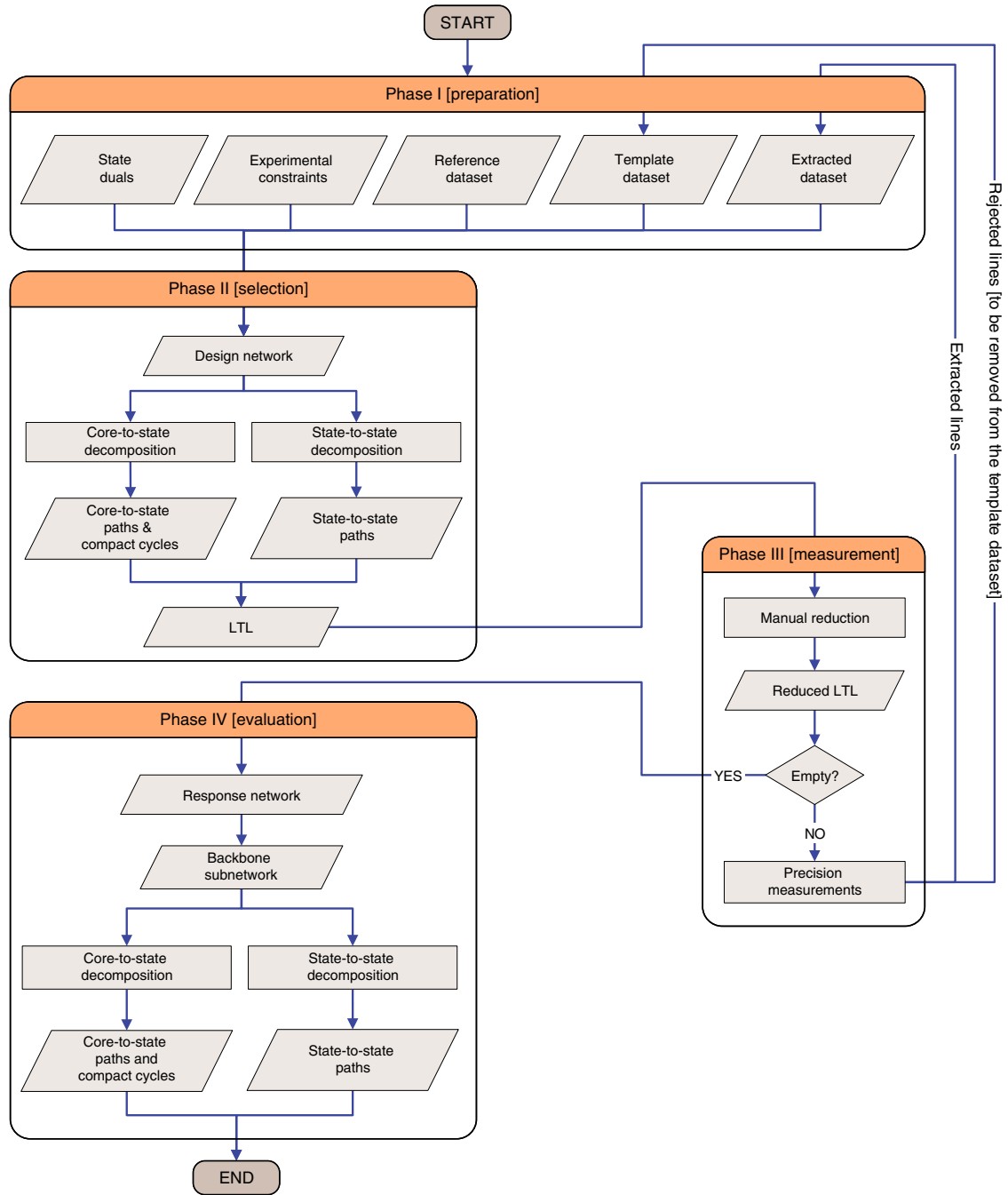

**Fig. 9 Flowchart of the SNAPS approach.** Boxes of this picture are explained in the "Theory" section.

Obviously, this weighting scheme can be arbitrarily modified and adapted to the experimental setup applied. The design network is utilized in two different ways. First, a shortest (core-to-state) path is built for every state from its core with a single execution of the Dijsktra procedure, and the basic cycles of the corresponding shortest-path forest (compact cycles) are determined (core-to-state decomposition). Second, the entire list of line-disjoint shortest (state-to-state) paths are composed for each state dual by successive application of the Dijkstra algorithm (state-to-state decomposition). During the repetition of Dijkstra's procedure, the transitions of the actual state-to-state path are temporarily ignored from the design network in each step (these ignored lines are reinstated right before the next state dual is treated). A core-to-state path from $(0\,0\,0)0_{0,0}$ to $(0\,0\,0)1_{1,1}$ and two state-to-state paths for the state dual $[(0\,0\,0)0_{0,0}, (0\,0\,0)1_{0,1}]$ can be found in Figs. 5 and 6, respectively.

At the end of this step, the core-to-state paths and the compact cycles (grouped by the components of the design network) are printed into a file and the state-to-state paths are stored in a separate file by state duals. In these files, paths and cycles

are indicated either as unexhausted or exhausted, depending on whether they contain any template transitions or not. The set of template lines involved in unexhausted paths and cycles (called the linked target linelist, LTL) is also collected into a third file for each component.

In phase III (measurement), a reduced set is created from the LTL by manually selecting the template transitions of those unexhausted paths and cycles which are of interest to the user. If there is at least one line in the reduced target linelist, it must be detected, resolved, and carefully analyzed via the precision-spectroscopy technique employed. Having completed the high-precision experiments, the extracted lines should be included in the extracted dataset, while the lines declined for some reason (for instance, due to incorrect primary line parameters or measurement difficulties) need to be deleted from the template dataset. Thereafter, phases II and III are repeated with the altered input until the reduced target linelist becomes empty.

Once the iteration involving phases II and III is stopped, a response network is assembled from the reference and the extracted datasets (phase IV—evaluation).

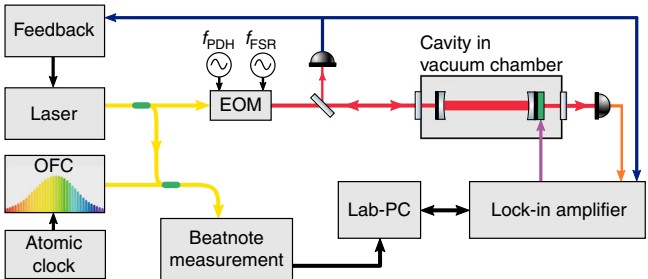

**Fig. 10 Schematic layout of the NICE-OHMS experimental setup.** The EOM (electro-optic modulator) is double modulated at both frequencies $f_{PDH}$ and $f_{FSR}$, which are 20 and 305 MHz, respectively, where the latter is exactly matched to the cavity-free spectral range. OFC, an optical frequency comb laser. For further details, see text.

The weights of the lines in the response subnetwork are initialized as the squared wavenumber uncertainties. Neglecting those reference transitions of the response subnetwork, which are not connected to extracted transitions (i.e., form separate islands), a backbone subnetwork is obtained. For this backbone subnetwork, the same decompositions are performed as for the design network of phase II, complemented with the evaluation of the energy differences and discrepancies concerning the output paths and cycles, respectively (see cases III–IV of Fig. 1). The relative energies of the rovibronic states (together with their uncertainties) are calculated from the core-to-state paths of the backbone subnetwork.

At the beginning of our measurement campaign on $H_2^{16}O$, the reference dataset consisted of the lines taken from refs. [31,37,39,42]. In search for line-disjoint paths from $(0\,0\,0)0_{0,0}$ to $(0\,0\,0)1_{0,1}$, virtual transitions (see the main text), constructed from the energy list taken from ref. [64], were also included to connect the *ortho* and *para* states (see also next subsection). For the template dataset, the line positions and their uncertainties were predicted from the empirical energies of the IUPAC database[22] of $H_2^{16}O$, while the line intensities and Einstein $A$ coefficients were taken from the first-principles linelist of ref. [64]. The experimental constraints guiding the SNAPS procedure of this study are as follows: [7000, 7350] cm$^{-1}$ for the wavenumbers, [10$^{-26}$, 10$^{-20}$] cm molecule$^{-1}$ for the intensities, and [10$^{-4}$, 10$^{2}$] s$^{-1}$ for the Einstein $A$ coefficients. The estimates for the primary line parameters of the template dataset proved to be sufficiently accurate to serve as design parameters for precision measurements performed with the NICE-OHMS setup.

**Linking *ortho* and *para* states.** The splittings of close-lying $(\nu_1\nu_2\nu_3)J_{J,0/1}$ states (i.e., the wavenumbers of the virtual lines) can be estimated as differences of first-principles energies. For these wavenumbers, the nuclear-motion computations[65,66] lead to sufficiently high accuracy (see Supplementary Note 3 and Supplementary Fig. 2), paving the way for the utilization of virtual lines to provide extremely precise lowest *ortho* energy determinations.

During the execution of the SNAPS procedure (see Methods), we took into account all the possible virtual transitions from ref. [64], with wavenumbers <10$^{-5}$ cm$^{-1}$, and searched for all the (shortest possible) line-disjoint paths defining the lowest *ortho* energy. This search yielded only two feasible line-disjoint paths within the experimental constraints, containing virtual lines with state pairs $(0\,4\,0)7_{7,0/1}$ and $(0\,4\,0)8_{8,0/1}$. The near-infrared lines of these paths (see Fig. 6) have been redetermined with the NICE-OHMS setup, while the wavenumbers of the virtual transitions have been benchmarked with variational nuclear-motion computations employing the GENIUSH[65,66] program system as well as several vibrational basis sets and potentials. The GENIUSH computations resulted in 4.26(18) × 10$^{-6}$ cm$^{-1}$ and 2.99(17) × 10$^{-6}$ cm$^{-1}$ for the wavenumbers of the $(040)7_{7,0} \leftarrow (040)7_{7,1}$ and $(040)8_{8,0} \leftarrow (040)8_{8,1}$ virtual lines, respectively. Further details on the GENIUSH computations, see Supplementary Note 4.

Utilizing the line positions and uncertainties of paths #1 and #2, together with the wavenumbers and the uncertainties of the virtual lines, the values of 23.794 361 54(59) and 23.794 361 41(71) cm$^{-1}$ are obtained for the lowest *ortho* energy, respectively. The accuracy of these determinations is dominantly determined by the excessive lengths of the underlying paths, and much less by the uncertainty contribution of the first-principles computations. These lowest *ortho* energy values are overall of slightly lower accuracy, but in excellent agreement with our effective Hamiltonian-based estimate.

**Experiment.** NICE-OHMS (see refs. [67,68]) is an ultrasensitive and ultrahigh-precision method of cavity-enhanced saturation spectroscopy, combining frequency modulation spectroscopy with cavity enhancement. In our NICE-OHMS setup, shown schematically in Fig. 10 and previously developed for precision metrology of hydrogen deuteride (HD)[47,48], we used an optical frequency comb as an external reference for long-term stabilization. In order to cancel short-time jitter, the external cavity diode laser is electronically locked via a

Pound–Drever–Hall (PDH) scheme to the spectroscopy cavity, which has a finesse of 150,000 and a length of 51 cm. Long-term stabilization of the cavity is achieved by beating the laser to an optical frequency comb laser disciplined by a cesium atomic clock and is further corrected by comparison with signals from the global positioning system. This advanced methodology gives rise to a frequency scale of sub-kHz accuracy. A double modulation scheme is applied via an electro-optic modulator for imposing the 20 and 305 MHz modulations for the PDH locking and for generating the spectroscopy sidebands, respectively, where the spectroscopy sidebands are carefully matched to the cavity-free spectral range, used in NICE-OHMS. While NICE-OHMS typically yields dispersive signals, in our detection scheme an additional layer of slow modulation is applied via dithering a piezo activated cavity mirror (at 405 Hz) to yield a derivative of the dispersive signal. Hence, our lines exhibit the shape of a Lamb dip as displayed in Fig. 4a.

The water vapor used for the measurements in this study originates from outgassing from the vacuum chamber walls, where water is the dominant component. Pressure levels are controlled by varying the pump speed to the turbo molecular pump to reach a steady state condition, where the pressure is monitored by a capacitance pressure gauge.

To improve the accuracy with which the line centers are measured, which is as low as 3–5 kHz, the broadening and shifting phenomena affecting the spectral lines detected in saturation in the NICE-OHMS setup were systematically investigated. The effect of the enhanced intracavity power gives rise to power broadening, and during the experiments, the intracavity power level was reduced and matched to the Einstein $A$ coefficient to avoid too strong broadening. The observed linewidths are on the order of 100 kHz.

As an example, the spectrum of the $(0\,0\,2)3_{3,0/1} \leftarrow (0\,0\,0)2_{2,1/0}$ *ortho–para* doublet was investigated under various conditions of intracavity saturation power. The values for the HWHM are plotted in Fig. 4b, clearly showing a power broadening effect. It was verified that power broadening does not lead to shifting of the central line position—no systematic effect was observed beyond the statistical uncertainty. Figure 4b also clearly demonstrates that the *ortho* line is slightly, but consistently, broader than the *para* line, which is attributed to the underlying unresolved hyperfine structure of *ortho*-$H_2^{16}O$. This effect of hyperfine structure does not lead to discernible asymmetries and a shift of the central line position, and hence does not compromise the accuracy of the determination of the *ortho* line centers.

A second systematic effect, caused by the influence of collisions and pressure, was also considered. While collisions and increased pressure do produce broadening of the lines, this is not of particular relevance for the present study. However, the collisional shift is a crucial issue because it is a limiting contribution in the measurement accuracy of the line centers. In Fig. 4c some typical pressure-dependence measurements are displayed for a number of lines. The explicit pressure dependence of the line positions has been determined only for a small fraction of the measured NICE-OHMS lines. In addition, due to the large amount of work required for the determination of the pressure shifts, the remaining transitions have been measured at a low pressure of around 0.1 Pa. Thus, based on the highest pressure-shift coefficient value (±20 kHz Pa$^{-1}$), a conservative pressure uncertainty of 2–3 kHz has been adopted for those lines for which no explicit pressure curves were measured.

Note that the relative values of the collisional shifts, and even the signs of the collisional shifts, for the newly determined saturated absorption lines do not agree with the self-collision shifts of the HITRAN database[1] for Doppler-broadened lines. This deviation may be attributed to the fact that in saturation spectroscopy only a partial velocity class is probed for the molecules, but a detailed analysis falls outside the scope of the present study.

Overall, a number of individual uncertainty factors have been assigned to all observed lines. First, the statistical uncertainties are calculated by taking different results of the fitting procedures for each single transition, and averaging over its multiple recordings. The pressure effects, mentioned above, have also been included in the line uncertainties. In addition, a conservative power shift uncertainty of 1 kHz has also been considered in the error budget. The final uncertainty of a particular line was estimated as the square root taken from the sum of squared individual uncertainty terms.

## Data availability

Data supporting the main findings of this work are available in the Supplementary Information files of this paper. The important experimental and calculated data of this study [list of old and new precision lines; high-accuracy predicted linelist; highly accurate relative rovibrational energies; output of the effective Hamiltonian fit; line-disjoint paths between $(0\,0\,0)0_{0,0}$ and $(0\,0\,0)1_{0,1}$; GENIUSH-based predictions for the wavenumbers of the $(0\,4\,0)7_{7,0} \leftarrow (0\,4\,0)7_{7,1}$ and $(0\,4\,0)8_{8,0} \leftarrow (0\,4\,0)8_{8,1}$ virtual lines; comparison of the various lowest *ortho* energy estimates; basic cycles built upon the new and old precision measurements; and comparison of the literature lines with our high-accuracy linelist] are provided as a Source Data file. The data underlying Figs. 2, 3, 5, 6, 7, and 8 can also be found in the Source Data file.

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

## Acknowledgements

This research received funding from LASERLAB-EUROPE (Grant No. 654148, European Union's Horizon 2020 research and innovation program). The work performed in Budapest received support from NKFIH (Grant No. K119658), from the grant VEKOP-2.3.2-16-2017-000, and from the ELTE Excellence Program (1783-3/2018/FEKUT-STRAT) of the Hungarian Ministry of Human Capacities (EMMI). W.U. acknowledges the European Research Council for an ERC Advanced Grant (Grant No. 670168). Further support was obtained from a NWO-FOM program (16MYSTP) and from the NWO Dutch Astrochemistry Network. The authors are grateful to Prof. Jonathan Tennyson for providing PES subroutines, as well as to Dr. Patrick Dupré, Dr. Csaba Fábri, and Dr. Tamás Szidarovszky for useful discussions.

## Author contributions

Conducted the experiments and resolved the lines: M.L.D., F.M.J.C., J.M.A.S., and E.J.S.; selected the linelist for the experiments and evaluated the resolved lines: R.T.; planned and created the draft of the figures: T.F., R.T., and F.M.J.C.; performed nuclear-motion computations and corrected the figures: I.S.; finalized the figures: F.M.J.C.; designed and supervised the research: A.G.C. and W.U.; all authors discussed the results and wrote the manuscript.

## Competing interests

The authors declare no competing interests.
