## [Peer Review File · Nature Communications]

REVIEWERS' COMMENTS:

Reviewer #1 (Remarks to the Author):

Manuscript#: NCOMMS-19-41757

Title: Spectroscopic-network-assisted precision spectroscopy: Application to water

The paper by Tobias et al. describes a holistic approach to network theory prediction and experimental validation of energy levels via ultra-high accuracy, precision molecular spectroscopy. The result is a set of ground-electronic-state energy levels of water at several hubs of the network, values which are reported herein with unparalleled breadth and accuracy. Furthermore, the completed network results in the highest accuracy measurement of the lowest ortho-water level energy.

The application of predictive networks to inform the targeted sub-Doppler spectroscopy of small molecules like water will have immediate impact, as current spectroscopic databases and radiative transfer codes (e.g., HITRAN) as well as Doppler-broadened comb-reference measurements [e.g., Sironneau and Hodges, JQSRT 152, 1-15 (2015)] are all less accurate by at least 3 orders of magnitude. Clearly, this comprehensive result immediately improves the line-by-line data for arguable the most important small molecule (water) and, in my opinion, will undoubtedly inspire further advances in interdisciplinary approaches at the cross-roads of computer science, first-principles quantum chemistry, and optical physics.

The statistical analysis appears consistent with best practices in reporting experimental uncertainty, and the details provided should allow a knowledgeable researcher with expertise in NICE-OHMS to reproduce the experimental results.

The accuracy and sensitivity of the NICE-OHMS experiments, as detailed in the methods section, merit acknowledgement in their own right as a noteworthy achievement. The application of sub-Doppler spectroscopy to >150 transitions is a painstaking task, and therefore having such experiments informed by a predictive network theory is powerful. I have several minor comments below, but overall rate the manuscript as one of the highest examples of interdisciplinary work in accurate spectroscopy in recent memory. Therefore, I believe that this paper will immediately achieve high impact -- and I strongly recommend its acceptance.

The empirical assignment of connections in the network appears to be a power approach to efficiently identifying transitions with local perturbations. This is currently beyond the grasp of standard spectroscopic models derived from an effective Hamiltonian. Would the authors comment on the predicted power of the SNAPS approach beyond the energy level connections measured?

As the authors correctly note, a similar number of sub-Doppler transitions have been measured for acetylene. However, the impact of a water line list goes well beyond the application of frequency makers in the telecommunications C-band (as is the case for acetylene). Water is the benchmark system for the study of advance line shapes in molecules, and water vapor also the largest contributor to the natural greenhouse effect [IPCC 2013]. It's abundance in the cosmos also necessitates improvements in models for radiative transfer in many environments.

Finally, I'm reminded of the AUTOFIT work of the Pate group [JMS 312, 13-21 (2015)] designed to confront the challenge of massive spectral congestion in CP-FTMW spectroscopy. There, initial brute-force approaches to identifying high-leverage transitions aided in the assignment of increasing complex microwave spectra. More recently Zaleski and Prozument [JCP 149, 104106 (2018)] have trained machine-learning algorithms to create an artificial neural network which can predict line-by-line spectra and assign their respective inertial parameters. Genetic algorithms have also been used in high-resolution electronic spectroscopy for 20 years by W. L. Meerts and co-workers [e.g. JCP 113, 7955 (2000), Can. J. Chem. 82, 804-819 (2004), JPCA 113, 5000-5012 (2009)]. If they so choose, the authors could briefly discuss their present application of advanced computer science approaches in relation to prior automated computer-aided routines in high-resolution spectra.

Adam J. Fleisher

National Institute of Standards and Technology

Gaithersburg, Maryland, U.S.A.

Reviewer #2 (Remarks to the Author):

Reviewer report:

on the manuscript by R.Tobias et al « Spectroscopic-network-assisted precision spectroscopy: application to water “

The paper reports very accurate measurements of water vapor near infrared line positions by cavity-enhanced optical heterodyne molecular spectroscopy combined with frequency modulation methods . This permitted measurements of 156 saturated Doppler-free absorption lines in the range 7000-7300 cm^{-1} with the intensities between 10^{-20} and 10^{-26} cm^{-1} molecule $^{-1}$.

The reported work represents a very significant effort in the extension of precise measurements of water vapor transitions towards the exceptional 10^{-7} cm^{-1} accuracy in line positions.

This was achieved by approaching nearly zero pressure limit in the infrared range experiments together with a careful choice of transitions for a consistent energy level determination. This in turn permitted generating new transitions via well known Ritz-Bohr-Einstein relation between the photon frequency and the upper and lower energy levels.

Information for accurate line positions is important for calibration purposes because water molecule is ubiquitous in various environments in terrestrial and astrophysical applications. They could also serve as reference values in the pressure line shifts studies.

The authors thoroughly discuss the sources of data as well as their validation and supply Supplementary files for observed and calculated data. The manuscript is in general clearly written.

I would recommend the authors to consider the following comments prior publication:

1. When describing the theoretical part of the paper (Ritz-SNAPS procedure) it would be fair to say that the Ritz principle had been already widely employed in the spectroscopic literature to determine rovibrational levels from observed transitions for various types of molecules accounting for the error propagations (for example *1-*5 below and refs therein as well as MARWELL series of the authors). The corresponding comments and citation are to be added. The original part at this part of the work is a consistent choice of target lines for laborious experimental measurements rather than a development of a new Ritz-based procedure for the data extension.

2. The new terminology “magic number” employed at page 14 instead of the conventional term “ortho-para energy splittings for $J=1$ ” would possibly be justified in case of a new physical content or

a new method of its determination. It is not the case. I would recommend using this old physically meaningful term for clarity. The corresponding value had been computed in many previous works, and here it comes out from a standard polynomial fit using the same very old effective model.

The question is whether this model (Watson effective rotational Hamiltonian) originally based on a perturbation theory in frame of Born-Oppenheimer approximation could really provide an accuracy of 10^{-7} cm⁻¹ ? In many previous works it was shown that this is not a best model for rotational levels of the water molecule (for example *5 below and refs therein) suffering from severe convergence and extrapolation issues.

3. This comment is linked to the previous one. The authors introduced 'artificial' transitions in the effective Hamiltonian fit that looks similar to a combination differences approach. A well-known risk of a polynomial fit with a large number of parameters concerns possible "fluctuations" for missing data points. The authors have to specify, how many energy differences of newly determined ground vibrational state (000) levels were used for the fit , and how many parameters of 14-th order Watson Hamiltonian were used ?

4. What about line intensities ? Was it possible improving these improve these important line parameters using reported high-precision measurements?

*1. J-M. Flaud, et al. Higher ro-vibrational levels of H₂O deduced from high resolution oxygen-hydrogen flame spectra between 2800–6200 cm⁻¹. Mol. Phys. 1976; 32:499–521.

*2. JKG Watson . The use of term-value fits in testing spectroscopic assignments. J. Mol. Spectrosc. 1994; 165:283–290.

*3. S.N. Mikhailenko et al. , Critical evaluation of measured rotation–vibration transitions and an experimental dataset of energy levels of HD18O / Journal of Quantitative Spectroscopy & Radiative Transfer 110 (2009) 597–608

*4. S.A. Tashkun et al. Critical evaluation of measured pure-rotation and rotation-vibration line positions and an experimental dataset of energy levels of ¹²C¹⁶O in X1S+ state / Journal of Quantitative Spectroscopy & Radiative Transfer 111 (2010) 1106–1116

*5. V.I. Starikov , et al . Description of vibration rotation energies of nonrigid triatomic molecules using the generating function method. Bending states of water molecule. J.Mol.Spectrosc., 1992,151, p130-147.

Reviewer #3 (Remarks to the Author):

This manuscript addresses the disconnect between the accuracy of the (mostly) Doppler-limited measurements used for spectroscopic databases (uncertainties often > 30 MHz even for water) with the extremely high accuracy now available with sub-Doppler and Doppler-Free experiments with frequency comb calibration (~3 kHz). Investigations of molecules at this high level of accuracy usually only result in a handful of transition frequencies and are unable to contribute much to databases composed of thousands of transitions over a broad frequency range. The authors argue that network-theory applied to spectroscopic networks (based on the Ritz principle) can determine a set of transitions which maximize the scientific utility of highly accurate measurements given the experimental constraints. They use a new network-theory based routine (SNAPS) to select 156 transitions of water within the sensitivity and frequency coverage of a NICE-OHMS instrument (7000-7350 cm⁻¹) and measure these transitions with 1.5-10 kHz uncertainties. From this limited data set, SNAPS predicted the frequency of 1219 transitions with uncertainties of ~10 kHz, including ~600 rotational transitions. To extend this to absolute energy levels, they connect the ortho and para manifolds using both nuclear motion calculations of ortho/para doublets which are linked with paths to the lowest ortho/para levels and with a fit of ground state rotational levels to an effective Hamiltonian.

This result is impressive, although measuring 156 Doppler-Free transitions with <10 kHz uncertainties by itself is not a trivial feat, it is especially significant that it could provide such an improvement to a large number of transitions at THz frequencies and energy levels relative to the ground rotational state. Spectroscopic databases are heavily utilized by a number of fields, including atmospheric science and astronomy communities, and it is valuable to have methods which can reveal systematic errors in past experiments and to improve the accuracy of available data. Absolute energy levels of molecules such as water are also useful as benchmarks for quantum chemistry and for models of hot exoplanetary atmospheres. The SNAPS routine will benefit other high-precision spectroscopy groups who are trying to perform spectroscopic surveys with the best possible efficiency. The manuscript is well written, with clear figures showing the working principles of SNAPS and spectroscopic networks. The statistical analysis for the error determination of the predicted

transitions and energy levels is done appropriately. I recommend the manuscript for publication after addressing a few minor issues and considering some suggested edits.

(Minor issues)

1. Introduction, paragraph 2, sentence 4: A NICE-OHMS setup is described as being disciplined to a Cs clock and to a GPS reference. It does not make sense to me for an instrument to use both frequency standards simultaneously and should either be described as “or” or one of the two.

2. Results and discussion, paragraph 2, sentence 2: It would be helpful to provide what uncertainty constitutes “former precision measurements”.

3. Transfer of measurement accuracy, paragraph 1, sentence 5: I do not directly see the connection between points a,b,c and being superior to an effective Hamiltonian. Points a and b (and in some cases c) are often true for predictions from molecular constants. The reason SNAPS would be superior to an effective Hamiltonian is that it only relies on the experimental values, and therefore will not suffer from issues like overfitting/choice of parameters. I suggest this sentence be revised to reflect this.

4. Results and discussion, paragraph 9, sentence 3: A typical path is shown, but as seen in Fig. 3 there are multiple paths to the (000) 1₁₁ level [through (101) 1₀₁ or (200) 1₁₁]. It would be helpful to include a statement regarding how the energy levels in redundant cases are determined. If a weighted average is taken or if the lowest uncertainty path is used.

5. Results and discussion, paragraph 10, sentence 3: It is unclear to me what the states were redefined with respect to. If possible, this sentence should be reworded for clarity.

6. Results and discussion, paragraph 10, sentence 3: Should the phrase “too weak target lines” instead be “two weak target lines” or “too weak of target lines”?

7. Experiment: Although a thorough description of the instrument is available in another publication, the manuscript would be more self-contained if even a short description were included either in the Experiment section or in the SM.

(Comments and suggestions)

1. Introduction, paragraph 2, sentence 4: This sentence is unclear. It could be misinterpreted as “the NICE-OHMS” setup being a general NICE-OHMS setup. Either rephrase it to read “In the NICE-OHMS setup used in this work...” or phrase it such that it reads as an example of the capability of NICE-OHMS.

2. NICE-OHMS precision spectroscopy of H₂O: In my opinion, it seems that the flow would be more logical to have paragraphs 2 and 3 switched. Paragraph 3 describes the experimental results related to paragraph 1. Paragraph 2 describes the confirmation of the uncertainty in paragraph 3 with SNAPS, and would logically follow descriptions of the experimental results.

3. Transfer of measurement accuracy, paragraph 1, sentence 4: It states that it “may have an accuracy characteristics of the NICE-OHMS setup”. Should there be reason to doubt this, or is it accounting for the accumulation of uncertainties from long paths?

4. Results and Discussion: It seems the closest thing to a concluding paragraph is paragraph 5 of “Magic number”. In my opinion, the manuscript would greatly benefit from a strong summary after “Frequency standards”.

5. Theory, paragraph 3, sentence 3: Was there a reason for weighting transitions by uncertainty instead of strength or a combination of factors? I could envision cases where it would be experimentally easier to measure one relatively uncertain value vs multiple more certain transitions, or measuring a stronger transition vs a weaker transition.

6. Theory paragraph 7: Typo, weights”

General notes: The manuscript has been carefully revised in response to the reviewers' comments and suggestions, responses to the comments are detailed below. Our reply is **highlighted in blue**, whereas the reviewers' specific comments are **in plain text**. All changes introduced in the manuscript are **also highlighted in blue**. We are greatly indebted to the reviewers for their suggestions, which have significantly improved our manuscript.

REVIEWERS' COMMENTS:

Reviewer #1 (Remarks to the Author): Manuscript #: NCOMMS-19-41757 Title: Spectroscopic-network-assisted precision spectroscopy: Application to water

The paper by Tobias et al. describes a holistic approach to network theory prediction and experimental validation of energy levels via ultra-high accuracy, precision molecular spectroscopy. The result is a set of ground-electronic-state energy levels of water at several hubs of the network, values which are reported herein with unparalleled breadth and accuracy. Furthermore, the completed network results in the highest accuracy measurement of the lowest ortho-water level energy.

The application of predictive networks to inform the targeted sub-Doppler spectroscopy of small molecules like water will have immediate impact, as current spectroscopic databases and radiative transfer codes (e.g., HITRAN) as well as Doppler-broadened comb-reference measurements [e.g., Sironneau and Hodges, JQSRT 152, 1-15 (2015)] are all less accurate by at least 3 orders of magnitude. Clearly, this comprehensive result immediately improves the line-by-line data for arguable the most important small molecule (water) and, in my opinion, will undoubtedly inspire further advances in interdisciplinary approaches at the cross-roads of computer science, first-principles quantum chemistry, and optical physics.

The statistical analysis appears consistent with best practices in reporting experimental uncertainty, and the details provide should allow a knowledgeable researcher with expertise in NICE-OHMS to reproduce the experimental results.

The accuracy and sensitivity of the NICE-OHMS experiments, as detailed in the methods section, merit acknowledgement in their own right as a noteworthy achievement. The application of sub-Doppler spectroscopy to >150 transitions is a painstaking task, and therefore having such experiments informed by a predictive network theory is powerful. I have several minor comments below, but overall rate the manuscript as one of the highest examples of interdisciplinary work in accurate spectroscopy in recent memory. Therefore, I believe that this paper will immediately achieve high impact – and I strongly recommend it’s acceptance.

The empirical assignment of connections in the network appears to be a power approach to efficiently identifying transitions with local perturbations. This is currently beyond the grasp of standard spectroscopic models derived from an effective Hamiltonian. Would the authors comment on the predicted power of the SNAPS approach beyond the energy level connections measured?

Response: The predictive power of the SNAPS procedure can be easily studied in basic cycles #8–11, #13, #18, #20–22, #24–26 (see sd9.txt of the zipped Source Data file), where only the new NICE-OHMS transitions are included. Considering basic cycle #9, the line $(021)8_{3,6} \leftarrow (000)7_{3,5}$ can be expressed with the combinations of transitions $(200)8_{0,8} \leftarrow (000)7_{3,5}$, $(200)8_{0,8} \leftarrow (000)7_{1,7}$, and $(021)8_{3,6} \leftarrow (000)7_{1,7}$. This combination provides a wavenumber of $7102.50574727(14) \text{ cm}^{-1}$ for $(021)8_{3,6} \leftarrow (000)7_{3,5}$, while the direct measurement leads to $7102.505747306(76) \text{ cm}^{-1}$, which deviates from the indirect prediction by only $3.60 \times 10^{-8} \text{ cm}^{-1}$ (corresponding to the discrepancy of basic cycle #9). Similarly good agreement (that is, prediction accuracy) can be observed in the other cycles. These examples suggest that the transition wavenumbers of our SNAPS-predicted lines should also be reproduced within their stated (a few times 10^{-7} cm^{-1}) uncertainties even via other experimental techniques.

As the authors correctly note, a similar number of sub-Doppler transitions have been measured for acetylene. However, the impact of a water line list goes well beyond the application of frequency makers in the telecommunications C-band (as is the case for acetylene). Water is the benchmark system for the study of advance line shapes in molecules, and water vapor also the largest contributor to the natural greenhouse effect [IPCC 2013]. It’s abundance in the cosmos also necessitates improvements in models for radiative transfer in many environments.

Response: Thank you for drawing our attention to these important applications. These research directions are now mentioned (with appropriate citations) in the Introduction.

Finally, I'm reminded of the AUTOFIT work of the Pate group [JMS 312, 13-21 (2015)] designed to confront the challenge of massive spectral congestion in CP-FTMW spectroscopy. There, initial brute-force approaches to identifying high-leverage transitions aided in the assignment of increasing complex microwave spectra. More recently Zaleski and Prozument [JCP 149, 104106 (2018)] have trained machine-learning algorithms to create an artificial neural network which can predict line-by-line spectra and assign their respective inertial parameters. Genetic algorithms have also been used in high-resolution electronic spectroscopy for 20 years by W. L. Meerts and co-workers [e.g., JCP 113, 7955 (2000), Can. J. Chem. 82, 804-819 (2004), JPCA 113, 5000-5012 (2009)]. If they so choose, the authors could briefly discuss their present application of advanced computer science approaches in relation to prior automated computer-aided routines in high-resolution spectra.

Response: Following the reviewer's useful suggestion, we added a sentence reviewing the relation of network theory to other computer-science approaches in the Introduction.

Adam J. Fleisher
National Institute of Standards and Technology
Gaithersburg, Maryland, U.S.A.

Reviewer #2 (Remarks to the Author): Reviewer report on the manuscript by R. Tobias et al. 'Spectroscopic-network-assisted precision spectroscopy: application to water'

The paper reports very accurate measurements of water vapor near infrared line positions by cavity-enhanced optical heterodyne molecular spectroscopy combined with frequency modulation methods. This permitted measurements of 156 saturated Doppler-free absorption lines in the range 7000–7300 cm^{-1} with the intensities between 10^{-20} and 10^{-26} cm molecule^{-1} .

The reported work represents a very significant effort in the extension of precise measurements of water vapor transitions towards the exceptional 10^{-7} cm^{-1} accuracy in line

positions. This was achieved by approaching nearly zero pressure limit in the infrared range experiments together with a careful choice of transitions for a consistent energy level determination. This in turn permitted generating new transitions via well known Ritz–Bohr–Einstein relation between the photon frequency and the upper and lower energy levels.

Information for accurate line positions is important for calibration purposes because water molecule is ubiquitous in various environments in terrestrial and astrophysical applications. They could also serve as reference values in the pressure line shifts studies. The authors thoroughly discuss the sources of data as well as their validation and supply Supplementary files for observed and calculated data. The manuscript is in general clearly written.

I would recommend the authors to consider the following comments prior publication:

1. When describing the theoretical part of the paper (Ritz–SNAPS procedure) it would be fair to say that the Ritz principle had been already widely employed in the spectroscopic literature to determine rovibrational levels from observed transitions for various types of molecules accounting for the error propagations (for example *1-*5 below and refs therein as well as MARVEL series of the authors). The corresponding comments and citation are to be added. The original part at this part of the work is a consistent choice of target lines for laborious experimental measurements rather than a development of a new Ritz-based procedure for the data extension.

Response: The reviewer is right in the sense that these methods implicitly use the paths and cycles of the input dataset. However, these techniques do not analyze the network structure; therefore, they are not able to reveal which transitions cause the inaccuracy of a particular energy level. In contrast, generation of paths and cycles enables us to find the lines strongly influencing the uncertainty of a given rovibronic level. (These important statements with the suggested references are now included in the Introduction.) Note also that although *2 in your reference list is devoted to the outlier detection in sets of rovibronic transitions, the method proposed by Watson relies only on a leave-one-out protocol, which is not satisfactory in most cases. We showed in a previous paper [J. Quant. Rad. Transfer, 203, 557–564 (2017)] that the explicit consideration of network cycles makes the outlier analysis much more efficient than accounting only for least-squares-based obs–calc deviations.

2. The new terminology ‘magic number’ employed at page 14 instead of the conventional term ‘ortho-para energy splittings for $J = 1$ ’ would possibly be justified in case of a new physical content or a new method of its determination. It is not the case. I would recommend using this old physically meaningful term for clarity. The corresponding value had been computed in many previous works, and here it comes out from a standard polynomial fit using the same very old effective model. The question is whether this model (Watson effective rotational Hamiltonian) originally based on a perturbation theory in frame of Born-Oppenheimer approximation could really provide an accuracy of 10^{-7} cm^{-1} ? In many previous works it was shown that this is not a best model for rotational levels of the water molecule (for example *5 below and refs therein) suffering from severe convergence and extrapolation issues.
- Response: The term ‘magic number’ has been replaced with ‘lowest *ortho* energy’ throughout the manuscript and the Supplementary Information. As to the use of the traditional Watson-type Hamiltonian for water, it is observed both by Toth [J. Opt. Soc. Am. B 8, 2236–2255 (1991)] and Matsushima et al. [J. Mol. Spectrosc. 235, 190–195 (2006)] that for lines involving low- J energy levels this model provides considerable accuracy. This observation coincides with our experience concerning the determination of the lowest *ortho* energy value of H_2^{16}O . Our new reference energy value for $(000)1_{0,1}$ has been adequately validated with our highly-accurate benchmark energies (having a maximum uncertainty of $u_{\text{max}} = 2.5 \times 10^{-7} \text{ cm}^{-1}$) of the lowest four para states $\{(000)1_{1,1}, (000)2_{0,2}, (000)2_{1,1}, \text{ and } (000)2_{2,0} - \text{verifying states}\}$. The verifying states acted as an external validation dataset for the effective Hamiltonian analysis (that is, their transitions were not fitted). The largest difference of the benchmark energies of the verifying states from the corresponding effective Hamiltonian estimates is only $1.3 \times 10^{-7} \text{ cm}^{-1}$, which is approximately half of u_{max} . Thus, we can expect that the $(000)1_{0,1}$ energy value coming from the effective Hamiltonian model has an uncertainty not significantly worse than u_{max} .
3. This comment is linked to the previous one. The authors introduced ‘artificial’ transitions in the effective Hamiltonian fit that looks similar to a combination differences approach. A well-known risk of a polynomial fit with a large number of parameters concerns possible ‘fluctuations’ for missing data points. The authors have to specify, how many energy differences of newly determined ground vibrational state

(000) levels were used for the fit, and how many parameters of 14-th order Watson Hamiltonian were used?

Response: Since our aim was only to determine the (000)_{1,1} energy value, we did not analyze the extrapolation capability of our effective Hamiltonian model for $J > 8$, where, as the reviewer correctly notes, Watson-type operators may suffer from serious convergence issues (mainly upon the increase of the K_a quantum number). Nevertheless, this pathological behavior appearing at higher J values does not affect the accuracy of the (000)_{1,1} energy (see our previous reasoning). The number of fitted lines and parameters were 111 and 44, respectively (see Supplementary Note 2 and sd5.txt in the zipped Source Data file); i.e., there are ≈ 2.5 times more points than parameters.

4. What about line intensities? Was it possible to improve these important line parameters using reported high-precision measurements? It should be noted that the current study is based on Doppler-free saturation measurements of spectral lines. Hence it is a study in non-linear optics. Although the line intensity in saturation follows somehow the transition dipole moment, it is not an obvious step to extract information of the Einstein- A coefficients, that is usually included in databases like HITRAN. Our study focuses fully on the quantum level structure of the water molecule, and this is done at high precision. Other important parameters, usually included in molecular databases, concern line broadening effects. We find in our study a different collisional effect than reported in HITRAN. This is illustrated in Fig 4c and discussed in Methods: crucial issue is that in saturation studies a subgroup in the molecular velocity distribution is probed.

- *1. J-M. Flaud et al., Higher ro-vibrational levels of H₂O deduced from high resolution oxygen-hydrogen flame spectra between 2800–6200 cm⁻¹. *Mol. Phys.* 32, 499–521 (1976).
- *2. JKG Watson, The use of term-value fits in testing spectroscopic assignments. *J. Mol. Spectrosc.* 16, 283–290 (1994).
- *3. S.N. Mikhailenko et al., Critical evaluation of measured rotation–vibration transitions and an experimental dataset of energy levels of HD¹⁸O. *J. Quant. Rad. Transfer* 110, 597–608 (2009)

- *4. S.A. Tashkun et al., Critical evaluation of measured pure-rotation and rotation-vibration line positions and an experimental dataset of energy levels of $^{12}\text{C}^{16}\text{O}$ in $X^1\Sigma^+$ state, *J. Quant. Rad. Transfer* 111, 1106–1116 (2010)
- *5. V.I. Starikov et al., Description of vibration rotation energies of nonrigid triatomic molecules using the generating function method. Bending states of water molecule. *J. Mol. Spectrosc.* 151, 130–147. (1992)

Reviewer #3 (Remarks to the Author):

This manuscript addresses the disconnect between the accuracy of the (mostly) Doppler-limited measurements used for spectroscopic databases (uncertainties often > 30 MHz even for water) with the extremely high accuracy now available with sub-Doppler and Doppler-Free experiments with frequency comb calibration (~ 3 kHz). Investigations of molecules at this high level of accuracy usually only result in a handful of transition frequencies and are unable to contribute much to databases composed of thousands of transitions over a broad frequency range. The authors argue that network-theory applied to spectroscopic networks (based on the Ritz principle) can determine a set of transitions which maximize the scientific utility of highly accurate measurements given the experimental constraints. They use a new network-theory based routine (SNAPS) to select 156 transitions of water within the sensitivity and frequency coverage of a NICE-OHMS instrument ($7000\text{--}7350\text{ cm}^{-1}$) and measure these transitions with 1.5-10 kHz uncertainties. From this limited data set, SNAPS predicted the frequency of 1219 transitions with uncertainties of ~ 10 kHz, including ~ 600 rotational transitions. To extend this to absolute energy levels, they connect the ortho and para manifolds using both nuclear motion calculations of ortho/para doublets which are linked with paths to the lowest ortho/para levels and with a fit of ground state rotational levels to an effective Hamiltonian.

This result is impressive, although measuring 156 Doppler-Free transitions with < 10 kHz uncertainties by itself is not a trivial feat, it is especially significant that it could provide such an improvement to a large number of transitions at THz frequencies and energy levels relative to the ground rotational state. Spectroscopic databases are heavily utilized by a number of fields, including atmospheric science and astronomy communities, and it is valuable to have methods which can reveal systematic errors in past experiments and to

improve the accuracy of available data. Absolute energy levels of molecules such as water are also useful as benchmarks for quantum chemistry and for models of hot exoplanetary atmospheres. The SNAPS routine will benefit other high-precision spectroscopy groups who are trying to perform spectroscopic surveys with the best possible efficiency. The manuscript is well written, with clear figures showing the working principles of SNAPS and spectroscopic networks. The statistical analysis for the error determination of the

predicted transitions and energy levels is done appropriately. I recommend the manuscript for publication after addressing a few minor issues and considering some suggested edits.

(Minor issues)

1. Introduction, paragraph 2, sentence 4: A NICE-OHMS setup is described as being disciplined to a Cs clock and to a GPS reference. It does not make sense to me for an instrument to use both frequency standards simultaneously and should either be described as ‘or’ or one of the two.

We have moved this statement from the introduction to the Methods section, where we have elaborated on the experiment, also explaining this issue. In fact the spectroscopy laser is locked to the optical cavity for short term stability, then locked also to the frequency comb (OFC), which itself is locked to an atomic clock (Cs-clock) for long-term stability and frequency determination, and further corrected by signals from the GPS system. This is mentioned in the text of Methods now.

2. Results and discussion, paragraph 2, sentence 2: It would be helpful to provide what uncertainty constitutes ‘former precision measurements’.

Response: The accuracy of former precision measurements is on the order of (some) kHz, which is now specified in the manuscript.

3. Transfer of measurement accuracy, paragraph 1, sentence 5: I do not directly see the connection between points a,b,c and being superior to an effective Hamiltonian. Points a and b (and in some cases c) are often true for predictions from molecular constants. The reason SNAPS would be superior to an effective Hamiltonian is that it only relies on the experimental values, and therefore will not suffer from issues like overfitting/choice of parameters. I suggest this sentence be revised to reflect this.

Response: The characterization of the SNAPS-based energy differences has been augmented with another point, where the pure experimental nature of the SNAPS method as well as the overfitting problem and the effects of different parameter choices are detailed.

4. Results and discussion, paragraph 9, sentence 3: A typical path is shown, but as seen in Fig. 3 there are multiple paths to the $(000)_{1,1}$ level [through $(101)_{1,0}$ or $(200)_{1,1}$]. It would be helpful to include a statement regarding how the energy levels in redundant cases are determined. If a weighted average is taken or if the lowest uncertainty path is used.

Response: The reviewer is correct in that the $(000)_{1,1}$ energy value can indeed be expressed with multiple ways from the network paths. However, if the wavenumber uncertainties are reliable, the different estimates deviate from each other only within the stated error bars. Thus, it is sufficient to use the shortest (minimal-uncertainty) path during the determination of a particular rovibronic energy. For example, the other path mentioned by the reviewer gives an energy value of $37.137\,125\,52(26)\text{ cm}^{-1}$ for $(000)_{1,1}$, while (in this exceptional case) the shortest-path-based prediction provides the same estimate with a slightly smaller uncertainty, $37.137\,125\,52(23)\text{ cm}^{-1}$ (the $37.137\,125\,2(23)\text{ cm}^{-1}$ value displayed in the caption to the ‘typical path’ figure had a transcription error, which has been corrected in the revised manuscript.) Since the practice how to choose paths for the determination of energy values was indeed not clear in the original manuscript, a clarifying sentence was added to the Methods/Theory section.

5. Results and discussion, paragraph 10, sentence 3: It is unclear to me what the states were redefined with respect to. If possible, this sentence should be reworded for clarity.

Response: In this sentence we have clarified that the redefinition of the energy values is based on paths formed by new and old precision lines.

6. Results and discussion, paragraph 10, sentence 3: Should the phrase ‘too weak target lines’ instead be ‘two weak target lines’ or ‘too weak of target lines’?

Response: The correct meaning is close to the second option of the reviewer. Corrected.

7. Experiment: Although a thorough description of the instrument is available in another publication, the manuscript would be more self-contained if even a short description were included either in the Experiment section or in the SM.

Response: A more extended discussion is now included in the methods (experiment section), providing important experimental details.

(Comments and suggestions)

1. Introduction, paragraph 2, sentence 4: This sentence is unclear. It could be misinterpreted as ‘the NICE-OHMS’ setup being a general NICE-OHMS setup. Either rephrase it to read ‘In the NICE-OHMS setup used in this work...’ or phrase it such that it reads as an example of the capability of NICE-OHMS.

We have taken out this statement on NICE-OHMS from the introduction on page 1, now introducing the NICE-OHMS technique from an experimental perspective later on, on page 5.

2. NICE-OHMS precision spectroscopy of H₂O: In my opinion, it seems that the flow would be more logical to have paragraphs 2 and 3 switched. Paragraph 3 describes the experimental results related to paragraph 1. Paragraph 2 describes the confirmation of the uncertainty in paragraph 3 with SNAPS, and would logically follow descriptions of the experimental results.

Response: The paragraphs mentioned by the reviewer have been interchanged in the manuscript. Also the flow of paragraphs in the introduction has been smoothed by taking out the specifics of NICE-OHMS - see statement above.

3. Transfer of measurement accuracy, paragraph 1, sentence 4: It states that it ‘may have an accuracy characteristics of the NICE-OHMS setup’. Should there be reason to doubt this, or is it accounting for the accumulation of uncertainties from long paths?

Response: The reviewer is right, long paths can increase the uncertainties of the derived energy differences, which is expressed by the ‘may’ modal verb. Fortunately, in our case, too large uncertainties coming from the error propagation have not been observed.

4. Results and Discussion: It seems the closest thing to a concluding paragraph is paragraph 5 of ‘Magic number’. In my opinion, the manuscript would greatly benefit from a strong summary after ‘Frequency standards’.

Response: Since papers in Nature Communications do not contain conclusions, we cannot include one in our manuscript. Some “concluding remarks” are collected in the last subsection of the Results and Discussion section.

5. Theory, paragraph 3, sentence 3: Was there a reason for weighting transitions by uncertainty instead of strength or a combination of factors? I could envision cases where it would be experimentally easier to measure one relatively uncertain value vs multiple more certain transitions, or measuring a stronger transition vs a weaker transition.

Response: We used the uncertainties of the template lines as weights, since the detection time could be significantly reduced if the initial positions were as accurate as possible. Obviously, the choice of the weighting scheme is arbitrary and can be optimized for the experimental setup applied. This arbitrariness of the weight function has been clarified in the Methods/Theory section.

6. Theory paragraph 7: Typo, 'weighths'

Response: Corrected.